# Exploring the Replication and Pathogenic Characteristics of Alpha, Delta, and Omicron Variants of SARS-CoV-2

**DOI:** 10.3390/ijms252312641

**Published:** 2024-11-25

**Authors:** Sakirul Khan, Takaaki Yahiro, Kazunori Kimitsuki, Takehiro Hashimoto, Keiko Matsuura, Shinji Yano, Kazuko Noguchi, Akane Sonezaki, Kaori Yoshizawa, Yoko Kumasako, Sheikh Mohammad Fazle Akbar, Akira Nishizono

**Affiliations:** 1Research Center for Global and Local Infectious Diseases, Oita University, Yufu, Oita 879-5593, Japan; 2Department of Microbiology, Faculty of Medicine, Oita University, Yufu, Oita 879-5593, Japan; 3Department of Advanced Medical Sciences, Faculty of Medicine, Oita University, Yufu, Oita 879-5593, Japan; 4Hospital Infection Control Center, Oita University Hospital, Yufu, Oita 879-5593, Japan; 5Department of Biomedicine, Faculty of Medicine, Oita University, Yufu, Oita 879-5593, Japan; 6Institute for Research Management, Oita University, Yufu, Oita 879-5593, Japan; 7Clinical Research Organization, Dhaka 1213, Bangladesh; 8Miyakawa Memorial Research Foundation, Tokyo 107-0062, Japan

**Keywords:** SARS-CoV-2, variants of concern, viral replication, nuclear localization, pathogenic potential

## Abstract

The variants of concern (VOCs) of SARS-CoV-2 have exhibited different phenotypic characteristics in clinical settings which are yet to be fully explored. This study aimed to characterize the viral replication features of major VOCs of SARS-CoV-2 and their association with pathogenicity. The Alpha, Delta, and Omicron variants of SARS-CoV-2 isolated from the COVID-19 patients in Japan were propagated in VeroE6/TMPRSS2 cells. The viral replication and pathological features were evaluated by laser and electron microscopy at different time points. The results revealed that the Delta variant dominantly infected the VeroE6/TMPRSS2 cells and formed increased syncytia compared to the Alpha and Omicron variants. Relatively large numbers of virions and increased immunoreactivities of the SARS-CoV-2 N-protein were detected in the endoplasmic reticulum and intracellular vesicles of Delta-infected cells. Interestingly, the N-protein and virions were detected in the nucleus of Delta-infected cells, while such properties were not observed in the case of Alpha and Omicron variants. In addition, early nuclear membrane damage followed by severe cellular damage was prominent in Delta-infected cells. A unique mutation (G215C) in the N-protein of the Delta variant is thought to be associated with severe cell damage. In conclusion, this study highlights the distinct replicative and pathogenic characteristics of the Delta variant of SARS-CoV-2 compared to the Alpha and Omicron variants, shedding light on the potential mechanisms underlying its increased pathogenicity.

## 1. Introduction

Severe acute respiratory syndrome coronavirus 2 (SARS-CoV-2) causes the coronavirus disease 19 (COVID-19), which was first reported in December 2019 in Wuhan, China. The COVID-19 pandemic represents an incredible pandemic involving billions of people. As of June 2024, over 775 million cases of COVID-19, with more than 7 million deaths, have been recorded worldwide [1]. However, circumstantial evidence and different observations indicate that both the number of COVID-19 cases and the deaths related to COVID-19 may be several-fold higher than reported [2,3,4]. The SARS-CoV-2 infection is ongoing and still poses one of the significant risks to public health worldwide due to the potential for phenotypic changes, transmissibility, and vaccine escape ability of the virus [5,6]. Since the beginning of the COVID-19 pandemic, numerous mutations of SARS-CoV2 have been identified [7]. Periodic viral genomic sequencing helps to detect new genetic variants circulating in communities. In addition, there remains the possibility of the emergence of new variants of SARS-CoV-2 or the re-emergence of previous variants with increased pathogenicity. Therefore, proper insights into the viral characteristics of SARS-CoV-2 variants are required to handle the situations in the coming days.

According to the World Health Organization (WHO), the Alpha, Delta, and Omicron variants of SARS-CoV-2 have been identified as dominant variants during the COVID-19 pandemic. The Alpha variant, first identified in the UK in late 2020, displayed higher transmissibility than the original Wuhan-Hu-1 strain and quickly spread globally. The Delta variant was detected in India in early 2021 and is more transmissible and linked to increased severity, leading to widespread surges and replacing Alpha as the dominant strain. Omicron, first reported in South Africa in late 2021, introduced significant mutations in the spike-protein, allowing it to spread rapidly. These variants of concern (VOCs) exhibit different phenotypic characteristics in clinical settings. The Delta variant of SARS-CoV-2 is known to cause more severe illness compared to earlier or later variants [8,9,10,11,12,13,14]. On the other hand, the Omicron variant appears to be more potent to transmit than other VOCs [12,13]. Available clinical data suggest that Omicron may cause a milder illness, and individuals infected with Omicron may have fewer hospitalizations compared to the Delta variant [9,11,14]. In fact, the Omicron variant is associated with lower disease severity than the Delta variant, with a case fatality ratio of 0.70% for Omicron versus 2.01% for Delta [10]. Although numerous clinical and experimental data indicate that the VOCs of SARS-CoV-2 have exhibited different pathological characteristics, some aspects of the cellular and molecular mechanisms underlying such differences are not fully explored.

SARS-CoV-2 primarily infects cells by binding its receptors and transmembrane protease on the cell surface. Once inside the cell, the virus releases its genetic material and utilizes the host cell’s machinery to replicate. SARS-CoV proteins are thought to interact with nuclear transport receptors and may allow the virus to enter the nucleus [15], although limited information is available about the nuclear localization of SARS-CoV-2. If any variants of SARS-CoV-2 gain access to the nucleus, it can potentially interfere with the cell’s normal functioning, leading to severe cell damage. Therefore, understanding the replication dynamics of VOCs of SARS-CoV-2 at the ultrastructural level is a crucial factor in the development of effective treatments and vaccines. Accordingly, by ultrastructural analysis, this study was performed to characterize the viral replication dynamics followed by the pathological features of major VOCs of SARS-CoV-2 in vitro. The results presented here indicate that the Delta variant of SARS-CoV-2 showed an increased ability to replicate in both cytoplasm and in the nucleus of infected cells, which is associated with significant cellular damage. This information would be of utmost importance to exploring why the Delta variant was more pathogenic, as well as tackling the VOCs of SARS-CoV-2.

## 2. Results

### 2.1. Delta Variant of SARS-CoV-2 Showed More Infectivity than the Alpha or Omicron Variant in VeroE6/TMPRSS2 Cells

The infection dynamics of the major VOCs of SARS-CoV-2 in the VeroE6/TMPRSS2 cells were confirmed by the recognition of the viral N-protein with immunofluorescence assay and Western blot (Figure 1). The immunoreactivities of the SARS-CoV-2 N-protein in the infected cells indicated that Omicron was less infective in the VeroE6/TMPRSS2 cells than the Alpha and Delta variants (Figure 1a–c). The immunoreactivities of the SARS-CoV-2 N-protein were detected in most of the cells at 48 h.p.i. when infected with the Alpha and Delta variants (Figure 1b). On the other hand, such immunoreactive activity was not observed in the case of the Omicron variant (Figure 1c). This finding was supported by the result of the Western blot of the SARS-CoV-2 N-protein (Figure 1d). Moreover, quite different morphology of cells was detected when they were infected with different VOCs of SARS-CoV-2. The Delta variant formed larger syncytia and plaque size than the other two VOCs of SARS-CoV-2 (Figure 1a–c), indicating that the Delta variant of SARS-CoV-2 may have more infectivity and pathogenicity in the VeroE6/TMPRSS2 cells.

### 2.2. Delta Variant of SARS-CoV-2 Showed More Replicative and Pathogenic Characteristics than the Alpha or Omicron Variant in VeroE6/TMPRSS2 Cells

To assess the viral replication, assembly, and pathological features of the major VOCs of SARS-CoV-2 at the cellular level, an electron microscopic analysis was performed (Figure 2, Figure 3, Figure 4, Figure 5, Figure 6 and Figure 7; Appendix A). The TEM analysis confirmed that the Delta-infected cells formed increased numbers of syncytia at the early infection stage than the Alpha or Omicron variant of SARS-CoV-2 (Appendix A) as observed in the immunofluorescence assay (Figure 1a–c). The TEM analysis also revealed that irrespective of VOCs, virions and the N-protein of SARS-CoV-2 were mainly assembled in the endoplasmic reticulum (ER) and intracellular vesicles of the infected cells during the early infections and increased virion numbers with time of infections (Figure 2, Figure 3 and Figure 4; Appendix A). However, the VOCs of SARS-CoV-2 had different replication potential in the VeroE6/TMPRSS2 cells. In the cases of the Delta variant of SARS-CoV-2, relatively large numbers of virion-containing ERs were detected in the cytoplasm at 24 h.p.i (Figure 2c). On the other hand, such distributions were not observed in the case of the Alpha or Omicron variant (Figure 2b,d). At 48 h.p.i, large numbers of virions were detected in the enlarged and swollen ERs of the Delta-infected cells, while relatively fewer virions were observed in the other VOCs, particularly the Omicron-infected cells (Figure 3d–f). Similarly to ERs, large numbers of virions were observed in the intracellular vesicles of the Delta-infected cells compared to those of the Alpha- or Omicron-infected cells (Figure 4a–f).

An immune electron microscopic analysis was conducted to confirm the replication characteristics of the N-protein of SARS-CoV-2 at the cellular level. The N-protein of SARS-CoV-2 was detected specifically, as no labeling was observed on the uninfected cells (Appendix A). However, the immunoreactivities of N-protein were dominantly detected in ERs (Figure 3a–c) and intracellular vesicles (Supplementary Appendix A) of the SARS-CoV-2-infected cells at 24 h.p.i. With increasing infection time, N-protein immunoreactivities decreased in the infected cells while the numbers of mature virions increased (Appendix A). As observed in the TEM, the immunogold labeling of the N-protein of SARS-CoV-2 revealed that the VOCs had different patterns of immunoreactivities in the infected cells. At 24 h.p.i., Delta infections exhibited strong positivity for the SARS-CoV-2 N-protein in ERs (Figure 3b) and intracellular vesicles (Appendix A). By contrast, in the other two VOCs, particularly the Omicron-infected cells, N-positive immunoreactivities were sporadically detected in ERs and intracellular vesicles (Figure 3a,c; Appendix Aa,c,d,f). Interestingly, the immunoreactivities of the SARS-CoV-2 N-protein were detected in the nucleus of the Delta-infected cells during early infection (Figure 5a,b). With increasing infection time, increased numbers of N-protein immunoreactivities and virions were observed in the nucleus when the cells were infected with the Delta variant (Figure 5c,d). The appearance of virions in the nucleus of the Delta-infected cells is similar to that of the virions observed in ERs and intracellular vesicles (Figure 3, Figure 4 and Figure 5). Moreover, the damage to the nucleus membrane of the Delta-infected cell was clearly observed at 48 h.p.i. (Figure 6). TEM and H&E analyses at 96 h.p.i. revealed that many cells were severely damaged when infected with the Delta variant of SRAS-CoV-2, while such damage was not seen in the other two VOCs (Figure 7; Appendix A). Collectively, compared to the Alpha and Omicron variants, the Delta variant of SARS-CoV-2 had more potential to replicate and assemble in the cytoplasm and nucleus of the TMPRSS2-expressing cells, and these features were associated with increased cellular damage.

### 2.3. Unique Mutations in N- and Spike-Protein May Link to Increased Infectivity and Pathogenicity of the Delta Variant of SARS-CoV-2

It has been reported that unique mutations in the N- and spike-protein of the Delta variant of SARS-CoV-2 allow the virus to become more pathogenic and infective, respectively [16,17]. To check such mutations, we performed a genomic analysis of the N- and spike-protein of the VOCs of SARS-CoV-2 that were used in immunofluorescence and TEM studies. Three unique mutations (D63G, G215C, and D376Y) were detected in the N-protein of the Delta variant when the sequences were compared with the Alpha, Omicron, and reference (Wuhan Hu-1) type of SARS-CoV-2 (Figure 8). Among these mutations, the G215C variation is thought to be linked to severe pathogenicity [16]. In addition, we found multiple mutations in spike-protein, notably P681R variation at a furin cleavage site of the Delta variant (Appendix A), which is thought to enhance replication via increased S1/S2 cleavage [17].

## 3. Discussion

The Delta variant of SARS-CoV-2 is known to be highly pathogenic and causes severe and prolonged illness compared to earlier and later VOCs [8,9,10,11,12,13,14,15]. On the other hand, the Omicron variant of SARS-CoV-2 causes a milder illness, and individuals infected with this variant may have fewer hospitalizations compared to other VOCs [12,13]. Various studies have documented the underlying virological features associated with such clinical discrepancies between VOCs of SARS-CoV-2 [18,19,20,21]. For example, the Omicron variant replicates faster than all the other SARS-CoV-2 variants in the bronchi but less efficiently in the lung parenchyma [18]. However, it is reported that Omicron is inefficient in using TMPRSS2 compared with wild-type SARS-CoV-2 and previous variants [21]. Also, the replication of the Omicron variant is significantly reduced in the respiratory tracts of hACE2 transgenic mice compared to the Delta variant, leading to markedly less severe lung pathology. In fact, the Omicron variant exhibits reduced production of viral particles throughout the respiratory system compared to earlier SARS-CoV-2 variants [18,21]. Although numerous aspects of the underlying virological features of the VOCs of SARS-CoV-2 are documented, many are yet to be fully explored. In this study, with ultrastructural analyses, we observed that the Delta variant of SARS-CoV-2 is more replicative in the cytoplasm during early infection than the Alpha or Omicron variant. The replication of the virus was also prominent in the nucleus of Delta-infected cells. These virological features of the Delta variant of SARS-CoV-2 are associated with the early destruction of the nucleus membrane followed by cell damage. Although it is not certain that the laboratory findings will completely mirror those in humans, our results suggest that the increased replication capacities in cytoplasm and nucleus may be associated with considerable cellular damage in alveoli tissue and subsequently develop severe disease after being infected with the Delta variant of SARS-CoV-2.

The cellular entry of SARS-CoV-2 depends on binding the viral spike-proteins to the receptors and cell proteases [22,23]. As this study used the cells expressing TMPRSS2, the cell surface protease that determines the SARS-CoV-2 entry pathway, it is assumed that the entry of the VOCs of SARS-CoV-2 into the cells mainly occurs by membrane fusion. However, the Delta variant of SARS-CoV-2 is found to be more infectious than the Alpha or Omicron variant, indicating that the Delta may have more affinity to the TMPRSS2 molecule. After entry into the cell, we found that the replication, assembly, and maturation processes of the VOCs of SARS-CoV-2 mostly occur in the cytoplasm of the infected cells, as reported by earlier studies [24,25,26,27]. Previous studies suggested that SARS-CoV is assembled in the ERs, where the N-protein binds to the genomic RNA and the M protein and forms virions [15,28,29]. In this study, irrespective of the VOCs of SARS-CoV-2, we found that the initial replication and accumulation of N-protein occurred in the ERs and intracellular vesicles where they formed virions, although the VOCs showed different replication capacities. For example, the Delta variant of SARS-CoV-2 replicated faster than the Alpha or Omicron variant in the ERs and intracellular vesicles. Moreover, the Delta variant of SARS-CoV-2 showed greater capacity for the formation of nucleocapsids.

Virus replication is blocked in enucleated cells and interferes with nuclear trafficking using inhibitors, which greatly reduces the infection [30,31], while the nucleus localization of the N-protein increases pathogenicity [15,32]. Despite the lack of information on SARS-CoV-2 replication into the nucleus, the present ultrastructural analysis revealed that the Delta variant of SARS-CoV-2 had the potential to replicate in the nucleus. In fact, the accumulation of SARS-CoV-2 N-protein was observed in the nucleus of the Delta-infected cells during early infection. In addition to the replication and accumulation of N-protein in the nucleus, we detected virions within the nucleus as observed in the cytoplasm. Also, we observed an association between virus replication in the nucleus and the early damage of the nucleus membrane of the Delta-infected cells. On the other hand, the present study found minimal nuclear localization and cellular damage potential for Alpha and Omicron variants. The N-protein of SARS-CoV-2 may play pivotal roles in the pathogenicity of certain variants through multiple mechanisms that extend beyond its ability to disrupt the host cell cycle [19,33,34]. Specifically, the N-protein has two primary functional domains contributing to viral pathogenicity, with varying impacts depending on the mutations present. First, the N-protein facilitates viral replication by supporting the assembly and packaging of RNA, leading to a rapid increase in viral load within host cells. This accelerated replication can produce large quantities of viruses, which may cause extensive tissue damage, particularly in the lungs and other affected organs [33]. Second, specific mutations in the N-protein enable it to enter the host cell nucleus, where it interferes with cell cycle regulation, potentially leading to cellular dysfunction and immune evasion [34]. Together, these mechanisms underscore the multifaceted role of the N-protein in viral virulence and emphasize that a complete understanding pathogenicity of the Delta variant requires the consideration of both its nuclear effects and its contribution to viral replication. With these realities, the present results indicate that the increased intracellular and intra-nuclear replication potential of the Delta variant of SARS-CoV-2 may lead to significant cell damage. Such divergent replicative and pathogenic features between the VOCs of SARS-CoV-2 in human alveolar cells will provide more insights into Delta-induced severe COVID-19.

The seroprevalence of SARS-CoV-2 has fluctuated with the emergence of major variants, including Alpha, Delta, and Omicron, each wave affecting different demographics and influencing the spread and immune landscape in diverse ways [35,36]. Seroprevalence data during the Alpha variant wave revealed significant increases in antibody levels among healthcare and frontline workers, while the highly transmissible Delta variant drove infection rates higher across broader age groups and geographic regions, increasing seroprevalence in communities with low vaccination coverage. The Omicron wave led to a rapid surge in cases worldwide, including high rates of breakthrough infections, resulting in widespread seroprevalence but with less severe disease outcomes. Across these variant waves, a portion of infected individuals developed long COVID, a condition characterized by persistent symptoms extending beyond the acute phase, which has pronounced psychological impacts [37,38]. Many individuals, particularly those who experienced severe disease or long COVID symptoms during the Delta wave, report increased levels of stress, depression, and social withdrawal, which have impacted their quality of life and ability to return to daily routines. Besides these, there are variations in the fatality ratio among the different variants of SARS-CoV-2 (Alpha: 2.62%, Delta: 2.01%, and Omicron: 0.70%) during the pandemic period [10]. These clinical data indicate that the Omicorn variant may have less pathogenic potential than earlier variants. Recent studies [8,9,10,11,12,13,14,15], including present ones, have revealed that the Delta variant of SARS-CoV-2 is more infectious and pathogenic than the Omicron variant. Compared to the original strain of SARS-CoV-2 (Wuhan-Hu-1), the Delta and Omicron variants contain dozens of mutations that may affect the viral phenotype [39]. While most mutations have been studied concerning the viral spike-protein, which mediates viral attachment and entry into the host cell, the N-protein is also crucial because it appears to handle virion packaging [33,40]. It has been reported that the N-protein of the Delta variant of SARS-CoV-2 has unique mutations that allow the virus to become more pathogenic. Worldwide database analysis of the VOCs of SARS-CoV-2 revealed that N-protein G215C mutation may be related to the severe pathogenicity of the Delta variant [16]. The Delta variant of SARS-CoV-2 used in the present study also showed G215C variation in the N-protein (Figure 8). It has also been reported that the P681R spike mutation increases the replication of the Delta variant by increasing the dissociation of S1 and S2 subunits at the furin cleavage site as reverting the P681R mutation to wild-type P681 significantly reduced the replication [17]. The present study also found P681R variation in the spike-protein of the Delta variant (Appendix A). Based on these findings, it is hypothesized that the P681R mutation in spike-protein might enhance the infectivity of the Delta variant by increasing S1/S2 cleavage and the G215C mutation in N-protein may be related to the severe pathogenicity of this variant.

This study has several limitations. While the present study primarily focused on the microscopic analysis of SARS-CoV-2 replication, it lacks well-defined statistical analysis based on quantitative data. Additionally, using kidney-origin VeroE6 cells may limit the generalizability of the findings to other cell types, particularly cells originating from the respiratory tract relevant to SARS-CoV-2 infection. Also, this study did not compare the replication and pathogenic features of the VOCs of SARS-CoV-2 with the Wuhan-Hu-1 strain, which restricts our understanding of how the Alpha, Delta, and Omicron variants differ in viral characteristics relative to the original strain. Furthermore, the study used only the Omicron lineage BA.1.1.2, limiting the applicability of our findings to other Omicron sublineages. As SARS-CoV-2 continues to evolve, it is possible that changes in its genomic composition could influence its replication and pathogenic features. Therefore, future research is needed to explore how the ongoing evolution of SARS-CoV-2 could impact viral characteristics. Despite these limitations, the results presented here provide important evidence that the Delta variant of SARS-CoV-2 exhibits greater replication and pathogenic potential than the Alpha and Omicron variants in VeroE6/TMPRSS2, likely due to its enhanced intracellular and intra-nuclear replication abilities.

## 4. Materials and Methods

### 4.1. Cells and SARS-CoV-2 Infection

The African green monkey kidney cells expressing transmembrane protease serine 2 (VeroE6/TMPRSS2) were used in this study. VeroE6/TMPRSS2 cells (JCRB1819), which are considered to have a high efficiency of SARS-CoV-2 infection [41], were purchased from the Japanese Collection of Research Bioresources (JCRB) Cell Bank (Osaka, Japan). The cells were maintained in Dulbecco’s modified Eagle’s medium (DMEM) containing 10% fetal bovine serum (FBS) and 1% penicillin-streptomycin (PS). The day before the infection, the cells were seeded into a 6-well plate and kept under 5% CO_2_ at 37 °C. All virus-related works were carried out in certified, high-containment biosafety level-3 facilities at the Department of Microbiology in Oita University Faculty of Medicine, Japan. The study procedures were approved by the ethical committee of Oita University School of Medicine, Oita, Japan (No. 1851).

The VOCs of SRAS-CoV-2 used in this study were isolated with nasal swab samples collected from COVID-19-confirmed patients in Japan during 2020–2022. The viral genome sequences confirmed the Alpha (lineage B.1.1.7; SARS-CoV-2/Human/Japan/TY-QHN001/2020; accession no. PQ380151), Delta (lineage AY.29; SARS-CoV-2/Human/Japan/OIT-0916/2021; accession no. PQ380152), and Omicron (lineage BA.1.1.2; SARS-CoV-2/Human/Japan/OIT-0211/2022; accession no. PQ380153) variants of SARS-CoV-2. The viruses were propagated into the VeroE6/TMPRSS2 cells at a dose of 1 × 10^5^ pfu/mL (in 20 µL per well). The virus was diluted in a culture medium at a multiplicity of infection (MOI) of 0.01 to establish a persistent infection. The cells were incubated for 24 h, 48 h, and 96 h under 5% CO_2_ at 37 °C. At the end of the incubation at each time point, the medium was removed, and the cells were fixed with fixatives, washed in phosphate-buffered saline (PBS), and processed for immunofluorescence, Western blot, electron microscopic, and histopathological analysis. For fixation, the infected and non-infected control (mock) cells were fixed in 2.0% glutaraldehyde (for ultrastructural analysis) or 10% formalin (for light microscopic and Western blot analysis).

### 4.2. Analysis of Infection Patterns of VOCs of SARS-Cov-2 by Immunofluorescence Staining and Western Blotting

The confirmation of the infection of SARS-CoV-2 in the VeroE6/TMPRSS2 cells was evaluated by the immunofluorescence and Western blot assays through the recognition of viral nucleocapsid (N-protein). Briefly, a paraffin block was prepared with formalin-fixed SARS-CoV-2-infected (Alpha, Delta, and Omicron variants; at 48 h after inoculation) cells. Cell sections were incubated overnight with rabbit anti-SARS-CoV-2 nucleocapsid (N-protein) antibody (1:1000; GeneTex, Inc., Irvine, CA, USA) at 4 °C. After washing in PBS, the sections were treated for 2 h at room temperature with an Alexa Fluor 568-conjugated goat anti-rabbit IgG (H + L) (1:1000; Invitrogen, Carlsbad, CA, USA), washed with PBS, mounted with Invitrogen™ ProLong™ Gold Antifade Mountant, and visualized using confocal LSM710 equipped with a 100× objective lens (Zeiss, Jena, Germany).

For the Western blot analysis, the VeroE6/TMPRSS2 cells infected with the VOCs of SARS-CoV-2 for 48 h were used. To quantify the SARS-CoV-2 nucleocapsid (N-protein) level in the infected cells, the collected cells were washed and lysed in lysis buffer and protease inhibitor cocktail (Nacalai Tesque Inc., Kyoto, Japan). Lysates were diluted with 2× sample buffer (100 mM Tris-HCl (pH 6.8), 4% SDS, 12% β-mercaptoethanol, 20% glycerol, and 0.05% bromophenol blue) and boiled for 2 min. Then, 10 μg samples were subjected to Western blot. For protein detection, rabbit anti-SARS-CoV-2 N-protein polyclonal antibody (1:1000; GeneTex, Inc., Irvine, CA, USA, GTX135357) and horseradish peroxidase (HRP)-conjugated goat anti-rabbit IgG (H + L) antibody (1:2000; Bio-Rad, Hercules, CA, USA, 1706515) were used. Bands were detected using the Clarity^TM^ Western enhanced chemiluminescence (ECL) Substrate (Bio-Rad), and images were acquired with an ImageQuant™ 800 (Amarsham).

### 4.3. Analysis of Viral Replication and Pathogenic Features of VOCs of SARS-CoV-2 by Electron Microscopy

For the analysis of viral replication and pathogenic features at the cellular level, transmission electron microscopy (TEM) was used. In this regard, the VOCs of SARS-CoV-2-infected and non-infected control (mock) cells were first fixed in 2.0% glutaraldehyde (0.2M, pH 7.2) overnight at 4 °C followed by being washed 3 times for 5 min in PBS and post-fixed for 30 min in 1% osmium tetroxide. The samples were taken at 24, 48, and 96 h post-inoculation (h.p.i.). The fixed cell samples were dehydrated through a graded series of ethanol (30, 50, 70, 85, 95, and 100%) and embedded in epoxy resin. The ultrathin sections (50~70nm) were stained with uranyl acetate and lead citrate.

For immunoelectron microscopy, sections were autoclaved (121 °C for 15 min in target retrieval solution) for antigen activation. After antigen retrieval, all the girds were immersed in a blocking reagent (Dako, Glostrup, Denmark) for 10 min to prevent the non-specific binding of the antibody. Next, the grids were incubated with anti-SARS-CoV-2 nucleocapsid (N-protein) antibody (1:1000; GeneTex, Inc.) overnight at 4 °C. After washing in TRIS-buffered saline (TBS) with 0.1% Tween 20, sections were incubated with 15 nm gold particles that had been conjugated with goat antibody against rabbit IgG (Amersham; Little Chalfont, Buckinghamshire, UK) for 30 min at room temperature. Followed by washing in TBS with 0.1% Tween 20, all the sections were counterstained with uranyl acetate and lead citrate in the usual manner. The ultrathin sections were observed using Jeol JEM 1011, FEI Titan, FEI Tecnai Sprit, and Hitachi HT 7800 electron microscopes (Tokyo, Japan).

### 4.4. Analysis of Pathological Features of VOCs of SARS-CoV-2 by Hematoxylin Staining

To investigate the cytopathological features of SARS CoV-2 VOCs, hematoxylin and eosin (H&E) staining was performed according to a standard protocol. Briefly, the paraffin block of formalin-fixed SARS-CoV-2-infected (Alpha, Delta, and Omicron variants at 96 h after inoculation) and non-infected (control) cells was stained with H&E to analyze pathological changes.

### 4.5. Analysis of Genomic Variations in VOCs of SARS-CoV-2

Viral RNA isolated from COVID-19-positive nasal samples were subjected to the whole genome sequencing of SARS-CoV-2 using a Next Generation Sequencer (Illumina iSeq100, San Diego, CA, USA). The DNA libraries were constructed with the Enhanced QIAseq Direct SARS-CoV-2 library preparation kit. The method begins with random-primed cDNA synthesis, followed by using multiplexed primer pools to prepare two pools of 225–275 bp SARS-CoV-2 overlapping amplicons (NEBNext ARTIC SARS-CoV-2 Primers, NIID ver. N6; Oxford Nanopore Technologies^®^, Oxford, UK). The two enriched pools from each sample were then combined into a single tube and cleaned with beads. All the enriched samples were amplified, sample-indexed with unique dual indices (UDIs), and subsequently cleaned with beads. The DNA libraries that had been amplified were quantified in a Quantus Fluorometer and thereby normalized before being uploaded into the Next Generation Sequencing machine.

The CLC genome workbench V12 software was used for genome analysis. Reference-based mapping was used to acquire the entire genome sequence of SARS-CoV-2. Multiple sequence alignment was performed to detect variation using the Molecular Evolutionary Genetics Analysis software (MEGA) version 11.

## 5. Conclusions

In conclusion, the present study found that the Delta variant of SARS-CoV-2 had more pathogenic potential than the Alpha or Omicron variant, which is associated with its increased intracellular and intra-nuclear replication ability. Although this work lacks the use of respiratory cells or the host response analyses after being infected with the VOCs of SARS-CoV-2, the replication specialties of the Delta variant of SARS-CoV-2 may be one of the underlying factors that are associated with severe clinical outcomes. Nevertheless, the results presented here underscore the importance of monitoring ultrastructural features and genomic mutations for effective variant surveillance as COVID-19 continues.

## Figures and Tables

**Figure 1 ijms-25-12641-f001:**
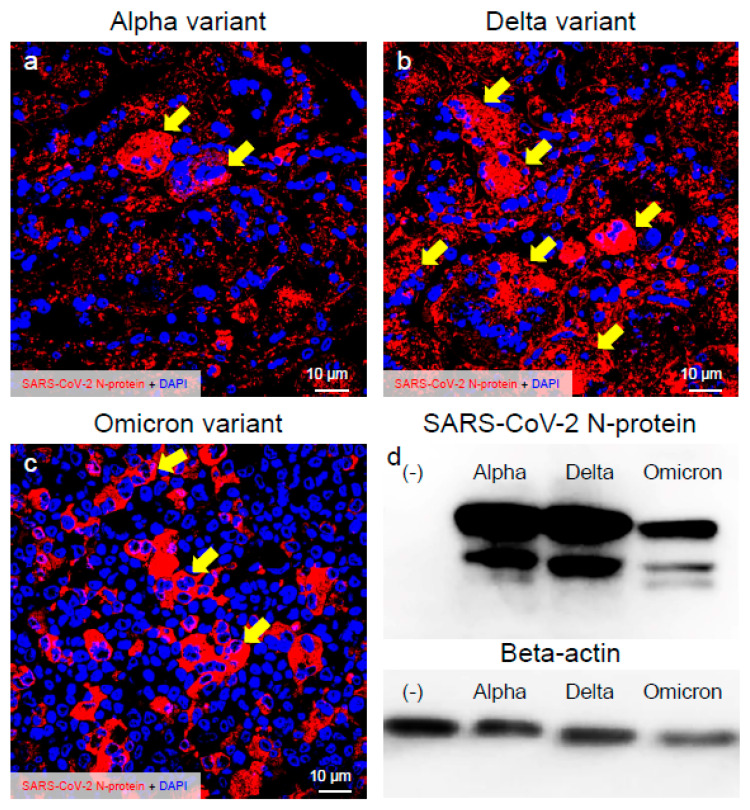
The Delta variant of severe acute respiratory syndrome coronavirus 2 (SARS-CoV-2) showed more infectivity than the Alpha or Omicron variant in the VeroE6/TMPRSS2 cells. (**a**–**c**) Infected cells at 48 h post-inoculation (h.p.i.) were stained with anti-SARS-CoV-2 nucleocapsid (N-protein) antibody. The representative confocal images indicate the infection patterns of the Alpha, Delta, and Omicron variants of SARS-CoV-2. Images show the immunoreactivity of the SARS-CoV-2 N-protein (red) and DAPI (blue). (**d**) SARS-CoV-2 N-protein was detected by immunoblot. Western blot analysis of extracts from mock and SARS-CoV-2-infected cells at 48 h.p.i. Beta-actin was used for internal control. Arrows indicate larger plaque.

**Figure 2 ijms-25-12641-f002:**
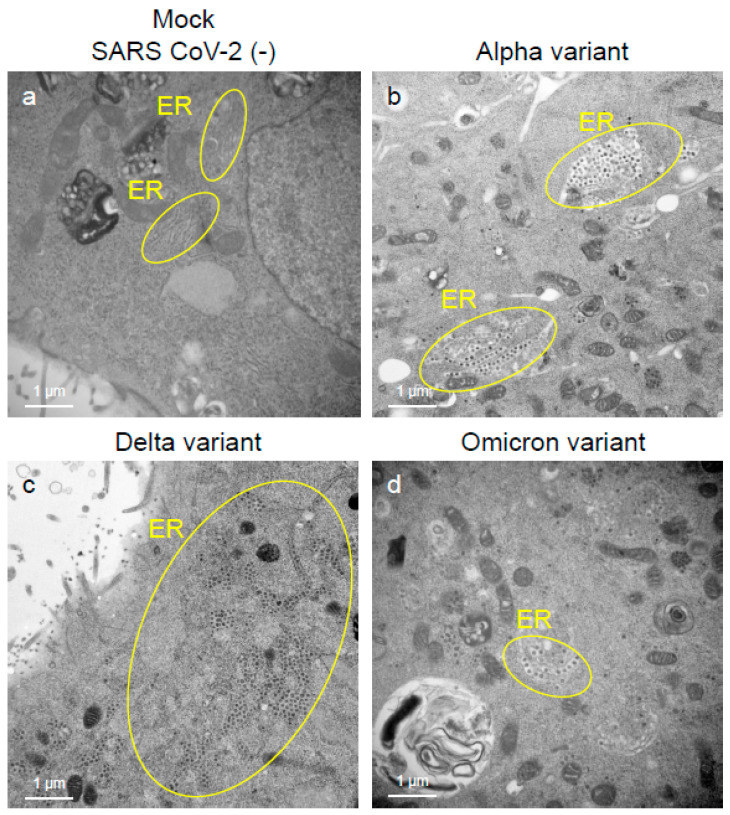
The Delta variant of severe acute respiratory syndrome coronavirus 2 (SARS-CoV-2) showed greater intracellular replicative potential in the VeroE6/TMPRSS2 cells than the Alpha or Omicron variant. (**a**–**d**) The representative transmission electron microscopic images indicate the distribution and accumulation patterns of the virions of the Alpha, Delta, and Omicron variants of SARS-CoV-2 in the endoplasmic reticulum (ER) at 24 h post-inoculation. Circles in (**b**–**d**) indicate the area of ERs where SARS-CoV-2 virions were detected.

**Figure 3 ijms-25-12641-f003:**
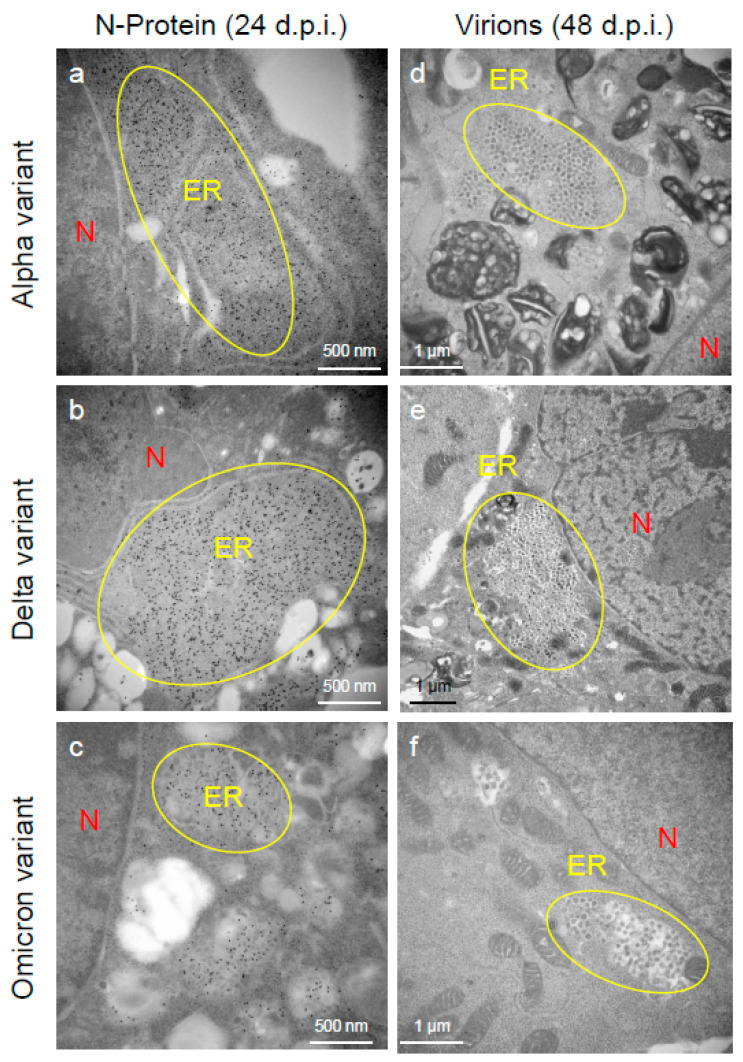
The Delta variant of severe acute respiratory syndrome coronavirus 2 (SARS-CoV-2) showed greater accumulation characteristics of the nucleocapsid (N-protein) and virions in the endoplasmic reticulum (ER) of the VeroE6/TMPRSS2 cells than the Alpha or Omicron variant. (**a**–**c**) The ultrathin sections of the infected cells at 24 h post-inoculation (h.p.i.) were stained with anti-SARS-CoV-2 N-protein antibody. The representative immunoelectron microscopic images indicate SARS-CoV-2 N-protein (gold labeling) distribution and accumulation patterns in the swollen ER. (**d**–**f**) The representative transmission electron microscopic images indicate the accumulation patterns of the virions of the Alpha, Delta, and Omicron variants of SARS-CoV-2 in the ER at 48 h.p.i. Circles indicate the swollen and enlarged ERs filled with N-protein of SARS-CoV-2 and/or virions. N: nucleus.

**Figure 4 ijms-25-12641-f004:**
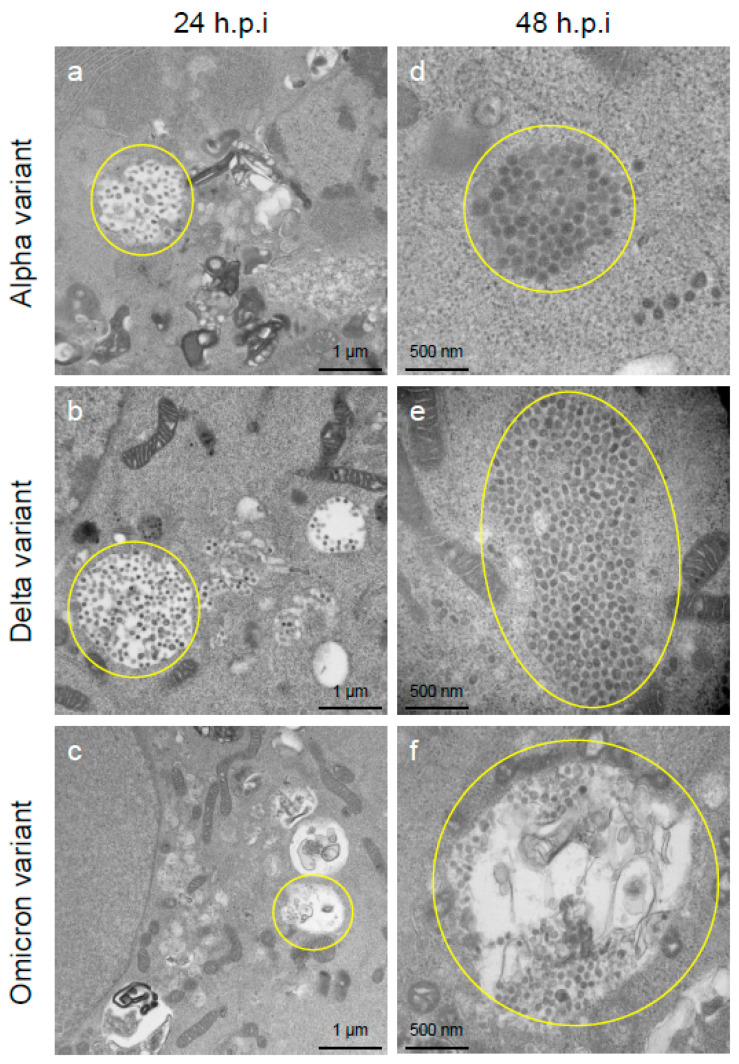
The Delta variant of severe acute respiratory syndrome coronavirus 2 (SARS-CoV-2) showed greater accumulation capacities of virions in the intracellular vesicles of the VeroE6/TMPRSS2 cells than the Alpha or Omicron variant. (**a**–**f**) The representative transmission electron microscopic images indicate the accumulation patterns of the virions of the Alpha, Delta, and Omicron variants of SARS-CoV-2 in large intracellular vesicles at 24 h post-inoculation (h.p.i.) and 48 h.p.i. The numerous SARS-CoV-2 virions of various sizes and spherical or multiform morphology accumulated in the large intracellular compartments (indicated by circles).

**Figure 5 ijms-25-12641-f005:**
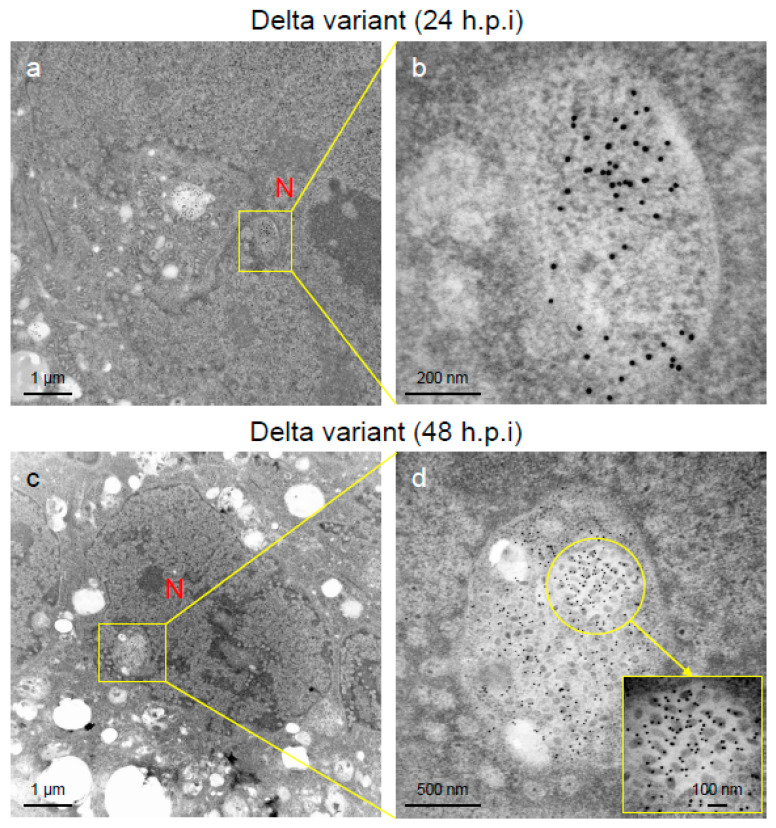
Nucleocapsid (N-protein) and virions of severe acute respiratory syndrome virus 2 (SARS-CoV-2) were detected in the nucleus of the Delta-infected VeroE6/TMPRSS2 cells. (**a**–**d**) The ultrathin sections of the infected cells at 24 h post-inoculation (h.p.i.) or 48 h.p.i were stained with anti-SARS-CoV-2 N-protein antibody. The representative immunoelectron microscopic images indicate the replication and accumulation patterns of SARS-CoV-2 N-protein (gold labeling) and virions of various sizes and multiform morphology in the nucleus of the Delta-infected cells. N: nucleus.

**Figure 6 ijms-25-12641-f006:**
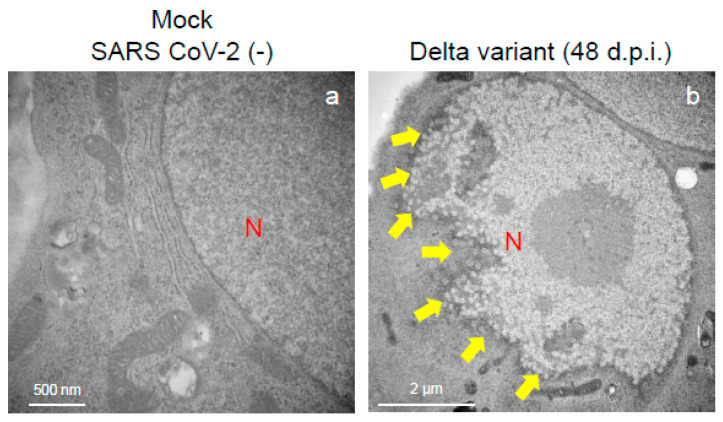
The Delta variant of severe acute respiratory syndrome coronavirus 2 (SARS-CoV-2) showed greater nuclear membrane damage potential in the VeroE6/TMPRSS2 cells. The representative transmission electron microscopy images indicate the intact nuclear membrane of mock-infected cells (**a**) and damaged nuclear membrane of Delta-infected cell (**b**). Arrows indicate the nuclear membrane damage at 48 h post-inoculation. N: nucleus.

**Figure 7 ijms-25-12641-f007:**
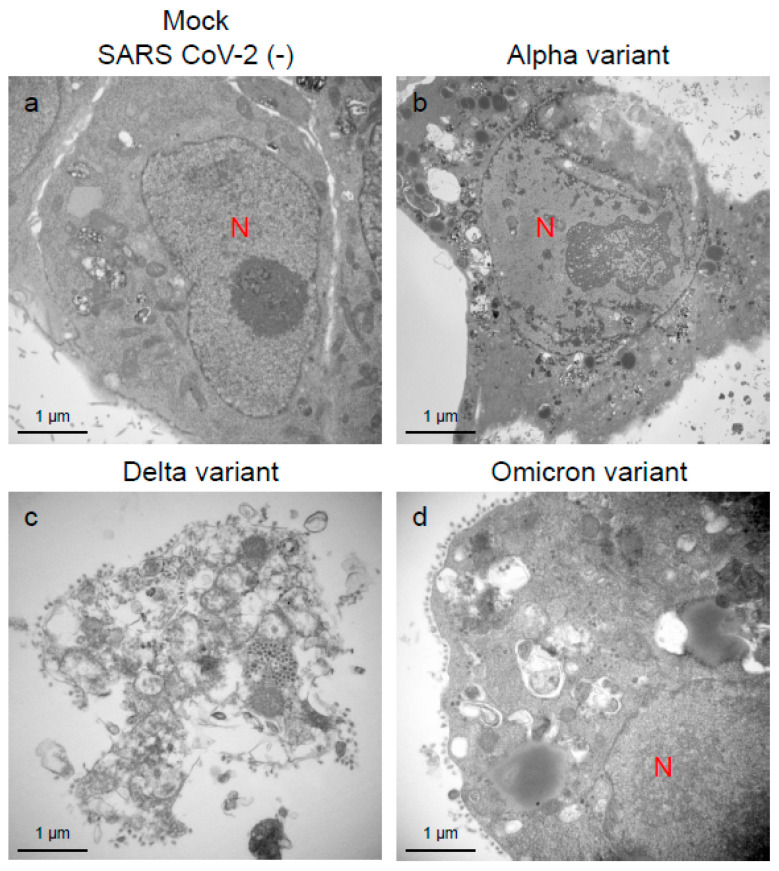
The Delta variant of severe acute respiratory syndrome coronavirus 2 (SARS-CoV-2) showed greater cellular damage potential in the VeroE6/TMPRSS2 cells than the Alpha or Omicron variant. (**a**–**d**) The representative transmission electron microscopic images indicate the cell damage patterns at 96 h post-inoculation of the Alpha, Delta, and Omicron variants of SARS-CoV-2. N: nucleus.

**Figure 8 ijms-25-12641-f008:**
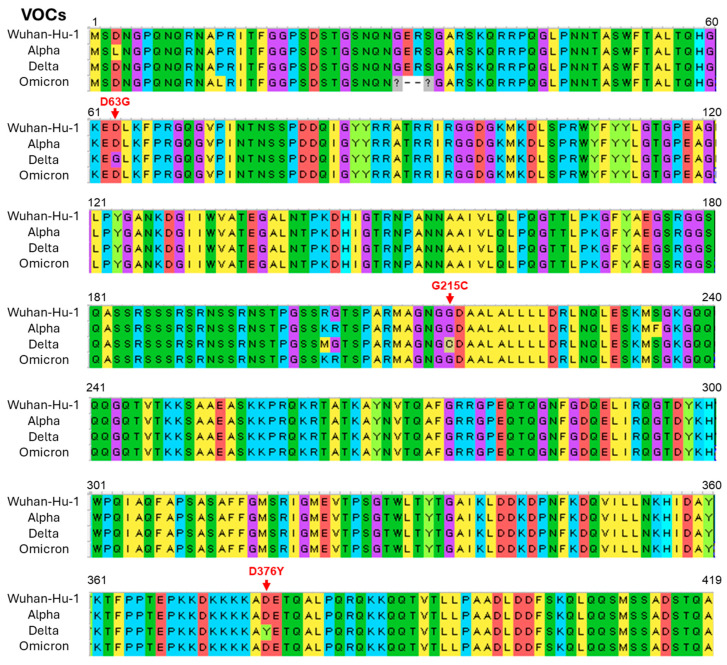
Unique mutations were detected in the nucleocapsid (N-protein) of the Delta variant of severe acute respiratory syndrome coronavirus 2 (SARS-CoV-2). Multiple sequence alignment of amino acids between the variants of concern (VOCs) and reference (Wuhan Hu-1) type of SARS CoV-2. Red arrows denote unique mutations in the Delta variant.

## Data Availability

The whole genome sequencing of Alpha (accession number: PQ380151), Delta (accession number: PQ380152), and Omicron (accession number: PQ380153) variants of SARS-CoV-2 have been deposited in GenBank of the National Center for Biotechnology Information (NCBI). The lead contact can provide any additional information required to reanalyze the data reported in this paper upon reasonable request.

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
