# Peer review of "Exploring the Replication and Pathogenic Characteristics of Alpha, Delta, and Omicron Variants of SARS-CoV-2"

_ijms, 2024, doi:10.3390/ijms252312641_

Round 1
Reviewer 1 Report
Comments and Suggestions for Authors
Estimated Authors,
I've been invited to review this very interesting paper, entitled "Exploring the Replication and Pathogenic Characteristics of Alpha, Delta, and Omicron Variants of SARS-CoV-2". In this study from the group led by Dr. Khan, Authors were able to provide an histological counterpart for the well-known clinical features of VOC alpha,delta and Omicron.
Notably, Authors were able to characterize Alpha as able to induce a sustained replication at very high rates, with high number of copies, earlier than other variants.
These results, as properly depicted by study Authors, contribute to explain the characteristics of the human infections from delta compared to alpha and particularly omicron.
In fact, Authors have provided a very well written and performed study, and the present Author has only marginal recommendations, including:
1) improve the quality of figure 8, that is scarcely readable
2) the study lacks of a "limits" section that could improve the overall quality of the paper. The main limit I'm guessing is that the study lacks of a well-defined statistical analysis reassuring the content of the paper. In fact, Authors sustain their statements on the analysis of histological images and TEM images; even though this limit does not impair the acceptance of the paper, Authors are welcome to acknowledge this potential issue; another limit is that omicron is a variant rapidly evolving and Authors should discuss whether the continuous evolution could impact or not on the biological features of the pathogen.
Author Response
Response to Reviewer # 1
Comments and Suggestions for Authors
I've been invited to review this very interesting paper, entitled "Exploring the Replication and Pathogenic Characteristics of Alpha, Delta, and Omicron Variants of SARS-CoV-2". In this study from the group led by Dr. Khan, Authors were able to provide an histological counterpart for the well-known clinical features of VOC alpha, delta and Omicron.
Notably, Authors were able to characterize Alpha as able to induce a sustained replication at very high rates, with high number of copies, earlier than other variants.
These results, as properly depicted by study Authors, contribute to explain the characteristics of the human infections from delta compared to alpha and particularly omicron.
Response: Thank you for your understanding and encouraging comments on our manuscript. We have reviewed your specific comments and made the necessary corrections. The insertions/alterations have been marked by “track change” in the revised text.
Specific comments
In fact, Authors have provided a very well written and performed study, and the present Author has only marginal recommendations, including:
1) improve the quality of figure 8, that is scarcely readable
Response: According to your suggestion, we have revised Fig. 8. Please see Fig. 8 in revised text [Page 9, between lines 209-210]
2) the study lacks of a "limits" section that could improve the overall quality of the paper. The main limit I'm guessing is that the study lacks of a well-defined statistical analysis reassuring the content of the paper. In fact, Authors sustain their statements on the analysis of histological images and TEM images; even though this limit does not impair the acceptance of the paper, Authors are welcome to acknowledge this potential issue; another limit is that omicron is a variant rapidly evolving and Authors should discuss whether the continuous evolution could impact or not on the biological features of the pathogen.
Response: According to your suggestion, we have inserted the limitations of our study in the revised text. Please see in revised text [Pages 11 & 12, lines 321-336]

Reviewer 2 Report
Comments and Suggestions for Authors
Khan and colleagues tried to characterize the viral replication features of major VOCs of SARS-CoV-2 and its association with pathogenicity.
The article has two challenges:
1) Topic COVID-19 seems to be less relevant than three years ago. Author know for sure that more than 200,000 articles on COVID-19 were already published. The article is about the past, not about the future or presence. I would say, it is historical . This is the reason why authors should very good explain to readers what the clinical relevance of their study for the presence and future is. I usually decline to review COVID-19 related articles since ne year as the topic is really over. But in this case I was interested based on the abstract.
2) Everything in this article is very difficult and will be understandable only for some less readers. I first thought , this is a kind of basic research, as a difference to original study.
Further issues:
3) In Methods authors should state a time period of experiments . Was is done in 20212022,2023? From when to when.
Images are very good.
Author Response
Response to Reviewer # 2
Comments and Suggestions for Authors
Khan and colleagues tried to characterize the viral replication features of major VOCs of SARS-CoV-2 and its association with pathogenicity.
Response: Thank you for your understanding and comments on our manuscript. We have reviewed your comments and made the necessary corrections. The insertions/alterations have been marked by “track change” in the revised text.
Specific comments
The article has two challenges:
1) Topic COVID-19 seems to be less relevant than three years ago. Author know for sure that more than 200,000 articles on COVID-19 were already published. The article is about the past, not about the future or presence. I would say, it is historical . This is the reason why authors should very good explain to readers what the clinical relevance of their study for the presence and future is. I usually decline to review COVID-19 related articles since ne year as the topic is really over. But in this case I was interested based on the abstract.
2) Everything in this article is very difficult and will be understandable only for some less readers. I first thought, this is a kind of basic research, as a difference to original study.
Response: Thank you for your assessment with criticism and appreciation. Based on your comments 2 & 3 with other reviewers’ suggestions, we have done major changes in the revised text particularly in Introduction and Discussion sections. Also, we provided the limitations of our study in the revised manuscript. All alterations have been marked by “track change” in the revised manuscript.
Further issues:
3) In Methods authors should state a time period of experiments . Was is done in 20212022,2023? From when to when.
Response: We have inserted this information in the revised text. Please see in revised text [Page 12, line 360]
Images are very good.
Response: Thank you for your understanding.

Reviewer 3 Report
Comments and Suggestions for Authors
Dear Authors,
thank you for this interesting and valid article of yours. Here are some small suggestions though:
1. In the first line write about 1st reported case of disease
2. In the introduction, write more about different types of virus, but highlight, that the ones you will focus on would be Omiron and Delta. Remember, there are planty of them, worth mentioning. In line 50, add the information on the year from which the data come
3. The figures nicely illustrate your findings.
4. In the discussion, add some psychosocial aspect of SARS-CoV2. Did it influence peoples lives? How did it influence respiratory system, including dimensions of airway? Did it influence peoples lives, ways of treatment (eg. doi:10.17219/dmp/157457)?
5. In the discussion, add the information on the seropositivity of the individuals, also referring to meditians (eg. DOI: 10.1186/s12879-023-08534-z). How does vaccination change the pandemic "patterns" and did it influence further life? Do not only focus on the "cells"
6. When you write about monkey cells as a use, explain why you picked these and also - did you need and had the ethical commitee approval? This statement is mandatory, even referring to cells' studies.
Thank you
Author Response
Response to Reviewer # 3
Comments and Suggestions for Authors
Dear Authors,
thank you for this interesting and valid article of yours. Here are some small suggestions though:
Response: Thank you for your understanding and comments on our manuscript. We have reviewed your comments and made the necessary corrections. The insertions/alterations have been marked by “track change” in the revised text.
Specific comments
- In the first line write about 1st reported case of disease
Response: According to your suggestion, this information has been inserted into the text. Please see in revised text [Page 1, lines 39-40]
- In the introduction, write more about different types of virus, but highlight, that the ones you will focus on would be Omiron and Delta. Remember, there are planty of them, worth mentioning. In line 50, add the information on the year from which the data come
Response: According to your suggestion, this information has been inserted into the text. Please see in revised text [Page 2, lines 54-59]
- The figures nicely illustrate your findings.
Response: Thank you for your understanding.
- In the discussion, add some psychosocial aspect of SARS-CoV2. Did it influence peoples lives? How did it influence respiratory system, including dimensions of airway? Did it influence peoples lives, ways of treatment (eg. doi:10.17219/dmp/157457)?
- In the discussion, add the information on the seropositivity of the individuals, also referring to meditians (eg. DOI: 10.1186/s12879-023-08534-z). How does vaccination change the pandemic "patterns" and did it influence further life? Do not only focus on the "cells"
Response: According to your suggestion in both comments 4 & 5, the information has been inserted into the text. Please see in revised text [Page 11, lines 285-299]
- When you write about monkey cells as a use, explain why you picked these and also - did you need and had the ethical commitee approval? This statement is mandatory, even referring to cells' studies.
Response: According to your suggestion, we have inserted the information in the text. Please see in revised text [Page 12, lines 348-350; 355-357].

Reviewer 4 Report
Comments and Suggestions for Authors
This is an interesting paper but there are some assumptions made that I don't think are correct.
1) The statements on Lines 210-211, 57-9 are simply wrong. The authors are saying this because they have not done sufficient literature search on this topic. There are papers addressing the same or similar concerns. The authors must track down all such papers and discuss what is the difference between their current paper and the papers from other labs. I know of at least two groups who have done similar experiments. One group have studied extensively M and N proteins using ai and experimental techniques.
https://www.nature.com/articles/s41586-022-04479-6
https://www.mdpi.com/1422-0067/25/14/7537
https://pubmed.ncbi.nlm.nih.gov/36291562/
https://www.nature.com/articles/s41586-022-04442-5
2) The experiment in the paper involves a narrow scope. For example, it uses only VERO-E6 cells and only three variants of SARS-CoV-2. Therefore, we cannot make sweeping claims without reminding the readers that there are limitations given the narrow scope of their experiment.
2a) This experiment involves the use of only VERO-E6 cell, which is a kidney cell. The authors observed that there was greater damage to cells in the presence of Delta and therefore Delta is more virulent. The authors cannot make such claim without explaining to the reader that this is just an extrapolation that is necessary given the limitation of the experiment. The authors need to explain that VERO-E6 is a kidney cell, not a respiratory cell, unlike CALU. The authors are extrapolating that such damage will occur in respiratory cells. Furthermore, only Delta and Alpha are used. Wuhan-Hu-1 is not used. This again requires extrapolation to come into any conclusion. All these have to be explained in the paper.
2b) Why was Wuhan-Hu-1 not used? This would have been a good reference virus since we know much more about it.
2c) The authors need to emphasise the limitations and potentials of their study. There are plenty of limitations as I can see but they seem to be swept under the carpet.
3) The papers mentions that Delta's greater virulence arises from its N ability to enter the nucleus to disrupt the cell cycle. The authors forget to mention that this is only one of N 's capability. CoVs have tow potential capabilities to cause virulence depending on which section of N is mutated. The first is the ability to replicate more rapidly arising from the role of N in assembly and packing the RNA. This can cause the reproduction of large quantity ofthe virus and can therefore be potentially damaging to the organ. The second is the ability to enter the nucleus to disrupt the cell cycle. The paper is not telling the whole story. If you want to make claim, you need to tell the whole story.
https://pubmed.ncbi.nlm.nih.gov/25105276/
https://www.mdpi.com/1422-0067/25/14/7537
https://virologyj.biomedcentral.com/articles/10.1186/s12985-023-01968-6
4) Do you have information on the CFR (case fatality ratio) of the various variants ? It could be used to back up what you are saying.
Author Response
Response to Reviewer # 4
Comments and Suggestions for Authors
This is an interesting paper but there are some assumptions made that I don't think are correct.
Response: Thank you for your understanding and comments on our manuscript. We have reviewed your comments and made the necessary corrections. The insertions/alterations have been marked by “track change” in the revised text.
Specific comments
1) The statements on Lines 210-211, 57-9 are simply wrong. The authors are saying this because they have not done sufficient literature search on this topic. There are papers addressing the same or similar concerns. The authors must track down all such papers and discuss what is the difference between their current paper and the papers from other labs. I know of at least two groups who have done similar experiments. One group have studied extensively M and N proteins using ai and experimental techniques.
https://www.nature.com/articles/s41586-022-04479-6
https://www.mdpi.com/1422-0067/25/14/7537
https://pubmed.ncbi.nlm.nih.gov/36291562/
https://www.nature.com/articles/s41586-022-04442-5
Response: Thank you for your comment and providing important publications. We have revised the statement and provided appropriate references in the text. Please see in revised text [Page 2, lines 67-69 and page 12, lines 220-229].
2) The experiment in the paper involves a narrow scope. For example, it uses only VERO-E6 cells and only three variants of SARS-CoV-2. Therefore, we cannot make sweeping claims without reminding the readers that there are limitations given the narrow scope of their experiment.
2a) This experiment involves the use of only VERO-E6 cell, which is a kidney cell. The authors observed that there was greater damage to cells in the presence of Delta and therefore Delta is more virulent. The authors cannot make such claim without explaining to the reader that this is just an extrapolation that is necessary given the limitation of the experiment. The authors need to explain that VERO-E6 is a kidney cell, not a respiratory cell, unlike CALU. The authors are extrapolating that such damage will occur in respiratory cells. Furthermore, only Delta and Alpha are used. Wuhan-Hu-1 is not used. This again requires extrapolation to come into any conclusion. All these have to be explained in the paper.
2b) Why was Wuhan-Hu-1 not used? This would have been a good reference virus since we know much more about it.
2c) The authors need to emphasise the limitations and potentials of their study. There are plenty of limitations as I can see but they seem to be swept under the carpet.
Response: Thank you for your suggestions. We understood the limitation of this study. According to your suggestion, we have inserted the limitations of our study in the revised text. Please see in revised text [Pages 11 & 12, lines 321-336]. We will consider your suggestions for our future studies.
3) The papers mentions that Delta's greater virulence arises from its N ability to enter the nucleus to disrupt the cell cycle. The authors forget to mention that this is only one of N 's capability. CoVs have tow potential capabilities to cause virulence depending on which section of N is mutated. The first is the ability to replicate more rapidly arising from the role of N in assembly and packing the RNA. This can cause the reproduction of large quantity ofthe virus and can therefore be potentially damaging to the organ. The second is the ability to enter the nucleus to disrupt the cell cycle. The paper is not telling the whole story. If you want to make claim, you need to tell the whole story.
https://pubmed.ncbi.nlm.nih.gov/25105276/
https://www.mdpi.com/1422-0067/25/14/7537
https://virologyj.biomedcentral.com/articles/10.1186/s12985-023-01968-6
Response: Thank you for your suggestions. We have revised the statement in the text. Please see in revised text [Pages 10 &11, lines 266-280].
4) Do you have information on the CFR (case fatality ratio) of the various variants ? It could be used to back up what you are saying.
Response: We have provided the information in the text. Please see in revised text [Page 2, line 66; Page 11, lines 299-300].

Round 2
Reviewer 2 Report
Comments and Suggestions for Authors
-
Author Response
Thank you for reviewing our manuscript and suggesting it be published in IJMS.
Reviewer 3 Report
Comments and Suggestions for Authors
Thank you for the corrections that improved your paper. Although the topic seems to be old, the paper is of a good quality now and could be published.
Author Response
Response to Reviewer # 3
Comments and Suggestions for Authors
Thank you for the corrections that improved your paper. Although the topic seems to be old, the paper is of a good quality now and could be published.
Response: Thank you for reviewing our manuscript and suggesting it be published in IJMS.
Reviewer 4 Report
Comments and Suggestions for Authors
1)Line 55 Wuhan is spelled wrongly.
2) Line 219-228. The authors did not mention the two papers already provided that showed that Omicron is producing less particles throughout the respiratory system. This is an important finding that must be mentioned.
Author Response
Response to Reviewer # 4
Comments and Suggestions for Authors
1)Line 55 Wuhan is spelled wrongly.
Response: We are sorry for our careless mistake. The spell has been corrected in revised text [Please see in Page 2, line 55]
2) Line 219-228. The authors did not mention the two papers already provided that showed that Omicron is producing less particles throughout the respiratory system. This is an important finding that must be mentioned.
Response: Thank you for your comments. According to your suggestions during the first round of revision, we have already inserted at least 2 references regarding the point you arisen about Omicron variant (Reference no. 17; Hui et al., 2022, Nature, 603(7902), 715–720 and Reference No. 20; Shuai et al., Nature, 603(7902), 693–699). There may be other suitable references but we are sorry for our limitations for not finding them. However, we inserted the statement regrading the ability of omicron to produce less particles in the respiratory system. [Please see in Page 210, lines 227-228]